# The Bioactive Core and Corona Synergism of Quantized Gold Enables Slowed Inflammation and Increased Tissue Regeneration in Wound Hypoxia

**DOI:** 10.3390/ijms21051699

**Published:** 2020-03-02

**Authors:** Lu-Chen Yeh, Shu-Ping Chen, Fang-Hsuean Liao, Te-Haw Wu, Yu-Ting Huang, Shu-Yi Lin

**Affiliations:** Institute of Biomedical Engineering and Nanomedicine, National Health Research Institutes 35 Keyan Road, Zhunan 35053, Taiwan; g9901301@gmail.com (L.-C.Y.); CSP@nhri.edu.tw (S.-P.C.); function@nhri.edu.tw (F.-H.L.); nadesico@nhri.edu.tw (T.-H.W.); smallbear@nhri.edu.tw (Y.-T.H.)

**Keywords:** quantized gold, endotoxin scavenging, oxygen providing, tissue regeneration

## Abstract

The progress of wound regeneration relies on inflammation management, while neovascular angiogenesis is a critical aspect of wound healing. In this study, the bioactive core and corona synergism of quantized gold (QG) were developed to simultaneously address these complicated issues, combining the abilities to eliminate endotoxins and provide oxygen. The QG was constructed from ultrasmall nanogold and a loosely packed amine-based corona via a simple process, but it could nonetheless eliminate endotoxins (a vital factor in inflammation also called lipopolysaccharides) and provide oxygen in situ for the remodeling of wound sites. Even while capturing endotoxins through electrostatic interactions, the catalytic active sites inside the nanogold could maintain its surface accessibility to automatically transform the overexpressed hydrogen peroxide in hypoxic wound regions into oxygen. Since the inflammatory stage is an essential stage of wound healing, the provision of endotoxin clearance by the outer organic corona of the QG could slow inflammation in a way that subsequently promoted two other important stages of wound bed healing, namely proliferation and remodeling. Relatedly, the efficacy of two forms of the QG, a liquid form and a dressing form, was demonstrated at wound sites in this study, with both forms promoting the development of granulation, including angiogenesis and collagen deposition. Thus, the simply fabricated dual function nanocomposite presented herein not only offers reduced batch-to-batch variation but also increased options for homecare treatments.

## 1. Introduction

Wound healing involves a complicated pathophysiological process that includes inflammation, proliferation, and remodeling. Chronic wounds normally take a long time to heal due to the fact that they remain in the inflammatory stage for too long, which interferes with the progression of the wound bed when going into the proliferation and remodeling phases. The infection of acute wounds with bacteria due to a lack of appropriate treatment is one of the main causes of chronic wound formation. Thus, the killing of bacteria is an essential aspect of initial wound treatment, as bacteria can trigger inflammation. At the same time, a form of debris called lipopolysaccharides (LPSs), also known as endotoxins, that can be released from the outer membrane of Gram-negative bacteria (GNB) are very toxic, and thus can act as a potent immune activator to slow down the process of wound healing. Therefore, it is a challenge to develop wound dressings that can effectively scavenge LPSs. LPSs are supramolecular and comprised of lipid A, core oligosaccharide, and O-antigen polysaccharide structures [1]. The lipid A structure of LPSs can bind with Toll-like receptor 4 (TLR 4) and then induce the activation of the NF-κB pathway and the expression of downstream proinflammatory cytokines to regulate immune responses [2,3,4]. Inflammation is not only related to innate immune responses, but is also involved in many severe diseases. Therefore, LPSs are a dangerous byproduct of bacteria insofar as they can cause an infected wound to exhibit worsening inflammation if the infection is not effectively controlled.

Inflammation management is just one of the essential aspects of wound care, and recent discoveries have revealed that oxygen plays a crucial role in the promotion of wound healing. Oxygen can regulate cellular activities involved in cellular respiration, metabolism, and aerobic glycolysis. Therefore, it acts not only as a substrate in metabolic processes, but also as a signaling molecule in biological processes [5]. In terms of wound care, sufficient oxygen is helpful for facilitating wound healing and tissue regeneration through such activities as bacterial defense, cell proliferation, angiogenesis, and collagen deposition [6,7]. To date, various approaches for increasing oxygen levels in wound sites have been introduced, including oxygen-delivering devices (e.g., hyperbaric oxygen therapy and topical oxygen therapy) and oxygen-generating materials. For example, hyperbaric oxygen therapy (HBOT) has been utilized as an oxygen delivery approach for the treatment of burns, diabetic foot ulcer wounds, and carbon monoxide poisoning. It is generally acknowledged, however, that HBOT may carry certain risks, including higher oxidative pressure or toxicity to organs [8]. Oxygen-generating materials such as calcium peroxide, magnesium peroxide, and fluorinated materials also come with substantial concerns related to toxic byproducts (such as salt compounds and overaccumulated reactive oxygen species (ROS)) [9]. As such, various additional oxygen-generating treatments for wound tissue have been developed and investigated in order to overcome these limitations. Hydrogen peroxide (H_2_O_2_) is one of the more common oxygen-releasing materials, in addition to being a major ROS in the hypoxic regions of injured tissue [10]. Therefore, if we are able to generate oxygen by utilizing the endogenous H_2_O_2_ in wound sites, this could offer a new strategy for wound treatment.

It is noteworthy, however, that the rapid consumption of H_2_O_2_ could also interfere with wound healing [11]. By promoting nanozyme development [12], the efficiency of artificial catalase in the decomposition of H_2_O_2_ into water and oxygen is lower than that of natural catalase, which could thus make the former a better choice compared to natural enzymes. Despite hundreds of materials having been developed into nanozymes, gold nanoparticles (nanogold) in particular have been widely used in a variety of biomedical applications and tools, such as biosensors, drug delivery systems, catalysts [13,14,15,16,17], and photothermal therapies, due to their biocompatibility. Again, the catalytic decomposition of H_2_O_2_ into water and oxygen on small-sized nanogold is more efficient than such decomposition on large-sized catalysts [17]. Not only does the core size matter in this regard, but also a well-ordered monolayer or a corona with a high packing density when entrapping the nanogold can pose big challenges in terms of blocking its intrinsic catalytic activity [18]. Polymer-encapsulated nanogold might thus be a good candidate due to its potential to simultaneously provide a quantized size of nanogold and a loosely packed outer corona. Thus, such a gold nanocomposite could be further designed as dual function quantized gold (QG) to bind with LPSs (the green rectangle in Figure 1, in which QG was encapsulated within a dendrimer), not influencing the catalytic function of its inner core. This suggests that the bioactive core and corona synergism of QG might assist in the management of wound beds by slowing down the overexpression of pro-informatory factors and reversing wound hypoxia. The concept is shown in Figure 1, which depicts the use of two forms of QG treatments, a liquid form and a dressing form, for tissue regeneration.

## 2. Results and Discussion

### 2.1. Morphology and Hydrophobic Properties of QG and GPSMA Nanofibers

The QG particles were mainly constructed using eight gold atoms entrapped within a dendrimer, and photoluminescence was emitted at a wavelength of 450 nm in aqueous solution (Figure 2A, the inset on the left side). Theoretically, the size of a single QG particle is smaller than 1 nm, which would make a single particle very difficult to observe using high-resolution transmission electron microscopy (HRTEM). Interestingly, TEM images showed a microcrystal structure when the QG solution was dropped onto a copper grid, which might have resulted from layer-by-layer stacking of single QG particles to form a thin layer (Figure 2A). Despite the alignment of the QG layers being random, the proximity distance of the gold atoms, which was in a range around 0.2884 nm [19], could still be estimated. As mentioned above, the optical and HRTEM measurements provided direct evidence to confirm the quantized size of the nanogold. Electrospinning is a process that is used to produce fibers ranging from nanometers to micrometers in thickness by ejecting a polymeric solution through voltage-driven equipment. Here, the PSMA nanofibers (10 wt% or 13 wt% of PSMA solution) were fabricated using electrospinning technology. The process used to form the PSMA nanofibers has the advantage that it is easy to generate a higher surface area by tuning the electrospinning parameters in order to obtain high grafting amounts of QG in the subsequent steps. As shown in Figure 2C,D, PSMA nanofibers with diameters of 660 nm could be produced with a 10 wt% polymeric solution, 15-kV voltage, 1-mL/h feeding rate, and 16-cm working distance.

After finishing the nanofiber preparation, QG was grafted onto the PSMA nanofibers via anhydride ring-opening reactions to form amide bonds. The successful grafting could be confirmed through a comparison of the FTIR spectra of the QG, PSMA, and GPSMA nanofibers, as shown in Figure 3A. This was because the QG was encapsulated by PAMAM dendrimers that showed two sets of amide peaks: amide I at 1647 cm^−1^ and amide II at 1557 cm^−1^ [20]. Otherwise, the absorption peaks of 1852 and 1780 cm^−1^ were attributable to the anhydride groups of the PSMA [21,22]. After GPSMA was formed, the characteristic absorption peak of the anhydride group almost disappeared, and new amide group peaks emerged. This phenomenon demonstrated that the QG was conjugated to the surface of the PSMA nanofibers. All of those samples had an alkyl group at 2862 to 2967 cm^−1^ assigned to a C-H bond stretching from the backbone of the PSMA. Additionally, the QG displayed fluorescence, with an emission wavelength at about 460 nm. Therefore, the fluorescence intensity of the QG in solution could be used to quantify the amounts of QG that were not grafted onto the PSMA nanofibers. A calibration curve was prepared using 0.1–1.6 mg/mL concentrations. Thus, each piece of PSMA nanofibers contained around 1.2 mg QG (Figure 3B). Moreover, QG grafting could change the surface wettability of PSMA as well. The contact angle of the unmodified PSMA nanofibers was 102°, which indicated that the intrinsic property of the PSMA was hydrophobic. However, the contact angle of the modified GPSMA was reduced to 0°, indicating the formation of hydrophilic nanofibers (Figure 3C). A macroscopic view of the GPSMA also revealed an obvious color change from white to yellow (Figure 3D).

### 2.2. Endotoxin-Binding Capability and Catalase-Like Activity of QG and GPSMA Nanofibers

Figure 4A shows that the intensity of the QG binding to LPSs was increased with increased QG loading amounts. Given the high binding affinity of QG and LPSs, this fact suggested that the PAMAM moiety of the QG provided a cationic anchoring region for the phosphate groups in the lipid A regions of the LPSs [23]. This phenomenon indicated that the QG could capture the anionic phosphate groups of the LPSs and then diminish the inflammation in wound beds by metabolizing the LPSs, which were captured by the gold nanoparticles. It should be emphasized that toxicity concerns about PAMAM dendrimers of a high generation could be eliminated while the hybrid nanocomposite was formed [24]; besides, the pack-loose of the organic layer (derived from dendrimers) had surface accessibility that kept the catalytic capability within the QG, which is an essential criterion in artificial enzyme mimetics (nanozymes). The recycling catalytic ability of the QG in the consumption of H_2_O_2_ was examined, and, as shown in Figure 4B, the turnover time of the catalytic reaction was within 2 h, after which the catalyzing efficiency remained. These results showed that the QG possessed long-term catalytic stability and a mild reaction rate compared to natural catalase, attributes that allowed it to avoid causing oxygen toxicity. Furthermore, it was also essential that the catalytic activity of GPSMA in consuming hydrogen peroxide be validated. As can be seen, the absorption intensity of H_2_O_2_ was dramatically decreased in the presence of QG or GPSMA (Figure 4C, blue and green lines), indicating that the QG maintained its catalytic activity even after being grafted onto PSMA nanofibers. Figure 4D summarizes the quantitative results of the measurements of H_2_O_2_ consumption. During the catalysis of H_2_O_2_, O_2_ was generated as a final molecule. Thus, we further aimed to observe the kinetic diagram of O_2_ generated within water during the decomposition of H_2_O_2_ through treatment with various matrixes. Accordingly, the amount of O_2_ produced in water was monitored using an oxygen microsensor. Here, catalase, a natural enzyme present in all aerobic organisms, was used as a positive control. It effectively decomposed H_2_O_2_ into water and O_2_, such that the concentration of O_2_ in the water reached a steady status fairly quickly (Figure 4E, red line). Interestingly, significantly higher oxygen concentrations were observed in the QG and GPSMA groups compared to the PSMA group, indicating that the gold nanozyme could provide oxygen via the consumption of H_2_O_2_ even if it was immobilized on PSMA. The postmodification of QG on the surface of PSMA was key to keeping the surface accessibility of QG for catalysis of the decomposition of hydrogen peroxide. These findings suggest that the QG can be used in various forms while maintaining its intrinsic catalytic properties as a nanozyme and providing oxygen to wound beds, a vital factor for wound healing.

### 2.3. Biocompatibility Study of GPSMA

To investigate the cytotoxicity of the QG, PSMA, and GPSMA, in vitro direct CCK8 assays were conducted to examine their biocompatibility with 1 × 10^5^ cells seeded per well. As shown in Figure 5A, the nontoxicity of these three samples was indicated by cell viability 1 day after being cocultured. Neither of the gold-containing materials had any statistically significant differences from the nanoparticle form or the nanofiber composite with regard to cell viability, but they did have higher cell viability than did the PSMA nanofibers. This phenomenon might have been contributed to by the intrinsic biocompatibility of nanogold. Thus, these results indicating low cytotoxicity confirmed that the QG and GPSMA are useful candidates for use in biomedicine applications such as wound dressing to improve wound repair and promote tissue regeneration.

### 2.4. GPSMA Diminishes NF-kB Activity In Vitro and Downregulates HIF-1α in Cells in Hypoxic Microenvironments

To investigate the effects of QG and GPSMA on injured tissue, we performed in vitro cell studies using human dermal fibroblasts (HDFs) and raw 264.7 cells. Here, a Western blot analysis was used to observe the expression of hypoxia-inducible factor-1 alpha (HIF-1α) and evaluate the activation of NF-κB. HDFs were cocultured with QG and GPSMA for up to 1 day in hypoxic incubators. As shown in Figure 5B, the HIF-1α level was downregulated in each treatment condition. These results were almost consistent with the results obtained from the catalase-like activity measurements. As mentioned above, endotoxin released from Gram-negative bacteria could activate the NF-κB pathway through the TLR 4 expressed in cells [25]. The cells were cultured and then treated with LPSs to trigger the inflammatory cascade reaction. As expected, the results showed that QG could slightly decrease the intracellular NF-κB levels (Figure 5C), which might change the self-assembly of LPSs in the presence of QG. It should be noted that LPS-based inflammation strength is known to result from different types of aggregates [26]. Comparatively, we found that GPSMA in the presence of LPSs showed higher NF-κB expression, suggesting that treatment with nanofibers could capture LPSs in the wound bed in a manner that could maintain the influence of LPSs for a period of time until a new GPSMA patch is applied, while still slowing down the inflammation. Taken together, the in vitro experiments demonstrated that QG could reduce the expression of HIF-1α and NF-κB proteins, which means that GPSMA (i.e., QG grafted on PSMA) can also reverse hypoxia and slow down the inflammation of cells in the wound bed.

### 2.5. Wound Closure after Treatment with Either QG Only or GPSMA

The creation of excisional skin wounds in mice is a common experimental method used to assess wound healing. Furthermore, endotoxins are a general stimulant for a series of wound healing models; therefore, full-thickness wounds in infected animal models injected with LPSs [27] were used to determine the effects of QG and GPSMA on local wound beds during the healing process. As shown in Figure 6A, the area of wound closure was altered in a time-dependent manner after injury, with wound healing in the QG group being significantly enhanced compared to that in the phosphate-buffered saline (PBS) and GPSMA groups on post-treatment day 3. On day 8, the wound recovery of the PBS group was approximately 90%, which was not significantly different from that of the QG group but was faster than the recovery rate for the GPSMA group (Figure 6B,C). However, even if the visual appearance of a wound indicates a faster closure rate, that does not mean that the wound in question is actually a well-repaired wound with well-developed granulation tissue as well as re-epithelialization.

### 2.6. The Granulation Development and Re-Epithelialization Impact on Wound Healing after Treatment with Either QG Only or GPSMA

Further assessments examined the possible influence of each treatment on the wound beds in each treatment group, with HE staining of the wound tissue being performed to observe the amount of new granulation tissue and re-epithelialization formation. First, as shown in Figure 7A,B, the average thickness of the wound granulation tissue in the QG group was about 700 μm on post-treatment day 8, which was much thicker than the average thicknesses in the other groups. Briefly, the degree of granulation could have been related to inflammation strength. Relatively speaking, the body excretions resulting from the application of QG alone may have assisted in LPS clearance in a way that quickly reduced inflammation in comparison to the application of GPSMA to wound sites. The appearance of inflammation on a wound bed is the first physiological symptom of the wound healing process, and excessive inflammation is a key factor in chronic wounds. Therefore, effective control of inflammation in the wound bed is a crucial and necessary aspect of effective wound healing treatments. An ELISA assay of an inflammatory cytokine from blood samples was used to examine the changes in inflammation during the wound healing process. As shown in Figure 7C, the results indicated that the amounts of IL-6 cytokine were slightly lower in mice treated with either QG or GPSMA compared to the control group (PBS) mice on day 1 after wounding. The results also indicated lesser degrees of NF-κB activation. Interestingly, the expression of IL-6 in the GPSMA group was higher than that in the QG group, suggesting that even as LPSs attached to the GPMSA via electrostatic interaction, there was still free lipid A on the surface of the nanofibers to induce inflammation. According to the ELISA results, QG can be utilized in two forms of wound dressing to reduce the amount of LPSs in local wound beds in order to shorten the inflammatory phase of wounds. Second, tissue hypoxia (i.e., low-oxygen tension) plays a key role in re-epithelialization, which is defined as the process of covering the epithelial surface of injured tissues. In the absence of re-epithelialization, a wound cannot be defined as having healed. Re-epithelialization involves the movement of keratinocytes at the wound edge. To migrate over the wound site, keratinocytes will change their normal phenotype and reorganize the cytoskeleton to enable the required migration, which is termed epithelial–mesenchymal transition (EMT) [28,29]. HIF-1α can act as one of the regulators of re-epithelialization; specifically, it has a positive effect during wound healing by increasing keratinocyte migration [30,31,32]. As mentioned previously, GPSMA nanofibers possess the catalase-like property to produce oxygen (Figure 4C–E), which can reverse the Hif-1α expression level of injured cells (also see Figure 5B). As demonstrated in Figure 7A, the neo-epithelium in the GPSMA group was the thinnest according to the H&E staining. The results thus strongly imply that gold-based wound dressing can provide oxygen locally, and they were also consistent with the results showing that the GPSMA group had relatively slow wound closure on day 8 (Figure 6C). To further study wound closure in the dermis layer, we observed the expression of α-smooth muscle actin (α-SMA), a key marker of myofibroblasts involved in wound contraction. Myofibroblasts are activated fibroblasts that are induced by the microenvironment and cytokines, such as HIF-1α and PDGF [33]. In the beginning of the tissue remodeling phase during the wound healing process, wound contraction contributes to wound closure. The process relies on the function of myofibroblasts to reduce the margins of the wound. After wound healing, the myofibroblasts physiologically undergo apoptosis to prevent excess contraction [34,35]. Compared to the GPSMA group, the injured mice treated with QG presented with a lower expression level of α-SMA protein on day 8 postinjury in Western blot measurements (Figure 7D). Combined with the results regarding the HIF-1α expression of HDFs in hypoxic environments and the wound closure measurements, those results make clear that wound repair in the QG group was faster than that in the GPSMA group. For the QG group, the wound contraction process was nearly completed, and the wound region had less α-SMA content at an earlier time point.

### 2.7. The Impacts of GPSMA on Skin Tissue Regeneration

Wound healing is a three-phase process consisting of the overlapping phases of inflammation, proliferation, and remodeling. It comprises monocyte infiltration, angiogenesis, re-epithelialization, extracellular matrix reconstruction, and wound closure. In order to investigate the impacts of GPSMA on the wound healing phases in a comprehensive manner, we focused on the impacts of QG and GPSMA on angiogenesis and collagen deposition, which are the key aspects of the proliferation and remodeling phases, respectively. ROS have the ability to regulate the formation of angiogenesis at a wound site. When a trauma in the skin is created, ROS production is initiated by hypoxia [36]. Furthermore, ROS mediate or recruit immune cells and platelets at the wound site [37], which in turn leads to the release of cytokines, one of which is vascular endothelial growth factor (VEGF), a potent protein produced by cells that is involved in the formation of blood vessels. Therefore, angiogenesis begins immediately after injury. However, while hypoxia can initiate neovascularization, it cannot sustain it. Rather, the process of angiogenesis can only proceed and be maintained in a microenvironment with a sufficient amount of oxygen [11]. As shown in Figure 8A, a Western blot analysis revealed that GPSMA upregulated the expression of CD31, a protein in endothelial cells, suggesting that it promoted angiogenesis by providing a microenvironment in which there were relatively higher amounts of oxygen than in the PBS and QG groups by transforming H_2_O_2_ into O_2_. The difference between the QG and the GPSMA was that the wound dressing form of the GPSMA could be retained long-term at a wound site. This result was consistent with Figure 7A, in which more blood vessels can be observed (white arrows). A therapeutic method that results in VEGF being added to a wound bed allows for improved neogenesis; however, VEGF also has a strong impact on scar formation, such that it could lead a wound bed to form fibrotic tissue [38]. As such, any vascular formation method that relies on adding to or otherwise mediating the level of VEGF needs to be utilized with care. Nonetheless, the results of this study indicate that GPSMA potentially has the ability to enhance angiogenesis while avoiding the risk of forming fibrotic tissue.

Not only does oxygen affect the development of blood vessels, but it also simultaneously supports the accumulation and deposition of collagen in the extracellular matrix. Oxygen-dependent prolyl 4-hydroxylase (P4H) is a hypoxia-inducible transcription factor. The function of P4H is to hydroxylate the proline of collagen, and it is a crucial factor for synthesizing [39] and providing thermal stability to collagen triple helices at temperatures of more than 37 °C [40,41]. The process of collagen hydroxylation is driven by oxygen tension; therefore, the generation of 4-hydroxyproline residues on collagen structures mostly occurs in normoxic environments [42]. A Western blot analysis was used in this study to examine the expressions of collagen I and collagen III, and Masson’s trichrome staining, in which aniline blue is normally used to color the collagen, was used to analyze collagen deposition. Figure 8B shows that there were differences between the QG- and PBS-treated wounds and the GPSMA-treated wounds, with the expressions of collagen I and collagen III in the GPSMA-treated wounds being higher than those in the PBS- and QG-treated wounds, suggesting that GPSMA can successfully promote collagen deposition in wound beds. The collagen structures formed in the dermal layer were observed using Masson’s trichrome staining on day 13 (Figure 8C). The collagen matrix was identified by the blue color. The GPSMA-treated wounds showed a higher intensity of the blue color, indicating a greater degree of collagen deposition, than did the wounds of the control group mice and QG group mice. On the basis of the above results, it can be concluded that the administration of QG could quickly metabolize LPSs, while the administration of GPSMA could transform the ROS into oxygen in the local injury area, meaning both treatments had positive effects in terms of promoting the development of granulation tissue.

## 3. Discussion

As we all know, an infected wound can be made to heal more rapidly if it is treated with antibiotics, which can effectively reduce the inflammation phase. This in turn causes the wound repair to be sped up and prevents the wound from becoming a chronic wound. However, antibiotic resistance is a growing public concern worldwide, and it can lead to systemic toxicity in some cases. On the basis of the in vitro and in vivo results of this study, we concluded that providing an appropriate supply of QG is the best method for shortening the inflammation phase of tissue, which in turn accelerates the next phase of the wound healing process. This shortening effect was made possible by the high affinity between QG and LPSs, which effectively sped up the excretion of the dangerous debris (i.e., LPSs) in the wound regions. As another form of gold matrix, we prepared QG grafted on nanofibers, which were then used as a wound dressing, and then compared the anti-inflammatory effects of QG and GPMSA on the wound beds. Although the anti-inflammatory effects of the GPSMA were a little less strong than those of the QG, the GPSMA could reduce the amount of LPSs within the wound bed through the replacement of the GPSMA nanofibers, to which the LPS became attached through electrostatic interactions (every two days). Interestingly, although both the QG and GPSMA had catalase-like activity, only the GPSMA could remain for a long time at the local injury area, so only it could enhance angiogenesis and collagen deposition by upregulating the decomposition rate of ROS in the hypoxic microenvironment. To our knowledge, it is impossible to use only one kind of wound dressing from the beginning to the end of an entire wound repair process, but it should be possible to choose the appropriate treatment depending on the state of a given wound. In this study, relatedly, we present different gold-containing wound dressings with good potential to assist in the healing of infected injury wounds in accordance with differing treatment purposes.

## 4. Materials and Methods

### 4.1. Fabrication and Characterization of PSMA Nanofibers and GPSMA

Poly(styrene-alt-maleic anhydride) (PSMA, Sigma, average Mw ~350,000) nanofibers were fabricated with electrospinning technology. Characterizations of the morphology, hydrophilicity, absorption of specific functional groups, and the amount of QG grafted on the PSMA nanofibers were investigated. In evaluating images of nonwoven mats, using an optical microscope is the simplest analytical method. The various pieces of equipment used for the electrospinning system consisted of a high-voltage power supply (Falco, FES-HV30), a syringe pump, and a collector. First, the PSMA polymeric solution was prepared in a cosolvent of THF/DMF (w/w = 3/7). The mixed solution was then placed into a 5-mL plastic syringe. Then, 18-mm-diameter round cover glasses were put on the collector, and PSMA nanofiber mats generated by different parameters were collected on the surfaces of the cover glasses. Before observing the fiber morphology, the collected nonwoven mats were dried in an oven overnight at 60 °C.

The synthesis of QG has previously been reported elsewhere [43,44]. The gold-modified PSMA nanofibers were fabricated using the following procedures. The electrospun PSMA nanofibers were soaked in 1 mL of QG solution (1.6 mg/mL) mixed with 10 μL of sodium hydrogen carbonate solution at 4 °C for 24 h. Next, the GPSMA was washed with deionized water three times, frozen at −20 °C in a freezer, and then freeze-dried. The PSMA was immersed in deionized water at 4 °C for 24 h, and the following processes were the same as those used for the GPSMA. The coating solution was collected, and the residual amounts of gold nanoclusters in the aqueous solution were measured using fluorescence spectrometry. An indirect method based on an internal calibration technique was used. Here, five different concentrations of QG aqueous solution were prepared, and a linear relationship was observed beyond the 0.1–1.6 mg/mL range (*R*^2^ = 0.995, *n* = 5).

The contact angle was used after the QG modification of the PSMA nanofiber mats: 4 μL of deionized water was carefully put in contact with the surface of the nonwoven fibers, and a photo was taken when each droplet on the surface was stabilized. Each sample was tested three times in 3 different spots. The specific chemical absorption bonds of the PSMA and GPSMA nanofibers were investigated using Fourier-transform infrared spectroscopy (FTIR, JASCO Inc., Easton, MD, USA). Samples were placed on a silicon wafer and analyzed within the wavenumber range 800–3500 cm^−1^, and spectra of the nanofibers were obtained at a resolution of 4 cm^−1^.

### 4.2. Endotoxin-Binding Assay

To assess the binding between the LPSs (Sigma, O111:B4 *Escherichia*
*coli*) and the QG, 30 μg/mL of LPSs in 0.1 M of Na_2_CO_3_ buffer containing 0.02 M EDTA was poured into a well plate and dried at 37 °C. Then, deionized water was used to wash the coated well plate before drying it again. Before measuring, the precoated plate was blocked with 1% BSA in phosphate-buffered saline (PBS) at 37 °C for 30 min and washed with 0.1% BSA in PBS. Varying concentrations of QG were added to the pretreated well plate and then incubated at 37 °C for 60 min. Then each well was washed with 0.1% BSA in PBS and deionized water three times. The fluorescence (Ex/Em at 390nm/460nm) was measured using an ELISA microplate reader (SpectraMax M2, Molecular Devices Inc., San Jose, CA, USA).

### 4.3. Catalase-Like Activity and Oxygen Generation of QG and GPSMA

The catalase-like activities of the QG and GPSMA were evaluated based on their consumption of H_2_O_2_. For the recycle test, QG was added to deionized water with 5 μM H_2_O_2_, after which a portion of the solution was mixed with 3,3′,5,5′-tetramethylbenzidine (TMB) and horseradish peroxidase (HRP) and then measured using UV–VIS analysis (ultraviolet–visible spectroscopy, VARIAN, Palo Alto, CA, USA). After 2 h, the same procedure was repeated. Then the same amount of H_2_O_2_ as the first time was added, and the test was repeated 4 times. In order to evaluate the catalase-like ability of the GPSMA, the samples (i.e., 1.0 mg of QG or one piece of GPSMA or PSMA) and H_2_O_2_ were added to deionized water, and then, after a while, TMB and HRP were added to the mixed solution. The residual H_2_O_2_ was measured using UV–VIS analysis. Then the related quantitative values were calculated using the value of H_2_O as a negative control and H_2_O_2_ as a positive control. An oxygen generation test with 0.1 mM H_2_O_2_ was performed using a commercial OXY-Meter sensor (Unisense, Aarhus N, Denmark). In preparing the control groups, H_2_O_2_ with catalase and H_2_O_2_ alone were used as the positive control and negative control, respectively. To monitor the oxygen generation, an electrode was inserted into the H_2_O_2_ solution, and the given sample (i.e., QG, GPSMA, or PSMA) was added to the solution. The experimental time was monitored for up to 30 min, and the test was performed at room temperature.

### 4.4. Cell Culture and Biocompatibility

A primary dermal fibroblast (HDF, ATCC PCS-201-012) cell line was cultured in fibroblast basal medium supplemented with fibroblast-growth-kit-low serum (ATCC PCS-201-041). The HDFs (4 × 10^4^ cells per well) were then cultured in 24-well plates at 37 °C in a humidified incubator with 20% O_2_ and 5% O_2_. After 1 day of subculture, the QG (1.2 mg/mL), GPSMA (one piece/mL), or PSMA (one piece/mL) was incubated with the HDFs for up to 1 day. After that, the mixed medium was removed, the HDFs were washed with Dulbecco’s Phosphate Buffered Saline (DPBS, Thermo Fisher Scientific Inc., Grand Island, NY, USA) three times, and then they were added to 250 μL of fresh medium mixed with CCK-8 solution for 2 h in an incubator. The biocompatibility was evaluated by measuring the absorbance at 490 nm using an ELISA reader.

### 4.5. Western Blot

For the HIF-1α expression assay, HDFs (10 × 10^4^ cells per well) were cultured in 6-well plates under normoxic conditions. After 3 days, the cells were cultured with mixed medium consisting of QG (1.2 mg/mL) or GPSMA (one piece/mL). Then the cells were incubated in an incubator with 1% O_2_ for up to 1 day. The HDFs were then harvested and centrifuged for 1 min at 4 °C. The experimental protocol that was followed was similar to the protocol described previously [43]. The cells were lysed with RIPA buffer with halt protease and phosphatase inhibitor cocktail. Whole-cell lysates containing equal amounts of protein were then loaded onto SDS-PAGE gels, separated by electrophoresis, transferred to PVDF, blocked with 5% BSA in TBST, and immunoblotted overnight at 4 °C with primary antibodies containing HIF-1α and beta-actin as a loading control. The membranes were then exposed to HRP and scanned using an Amersham Imager 600 (GE Healthcare UK Limited, Buckinghamshire, UK). To evaluate the inhibition of NF-kB (Cell Signaling, #3033) expression, murine raw 264.7 macrophages were seeded in a 6-cm dish and cultured with complete Roswell Park Memorial Institute (RPMI)-1640 medium (Thermo Fisher Scientific Inc., Grand Island, NY, USA) containing with 10% heat-inactivated FBS, 100 U/mL penicillin, and 100 μg/mL streptomycin. Then the processes that were followed were similar to those described above, except that the primary antibody of phosphor-NH-kB p65 (Ser536, Cell Signaling Technology, Inc., Danvers, MA, USA) was used.

### 4.6. Wound Healing Observation

Male 4-week-old C57BL/6JNarl mice were obtained from the National Laboratory Animal Center (Taipei, Taiwan). The animal procedures (the approval code: NHRI-IACUC-105138-M2-A September 2018) used followed published guidelines approved by the Institutional Animal Care and Use Committee of the National Health Research Institute (NHRI). The animals were raised in plastic cages with moderated humidity and temperature at the Laboratory Animal Center of the NHRI. An infected animal model was created by subcutaneously injecting each mouse with 25 μg LPSs [27]. Then the back of each mouse was shaved under anesthesia, and a 5-mm disposable biopsy punch was used to create two full-thickness wounds on the dorsal surface of the given mouse. For the QG group, 5 μL of QG solution (80 μg/μL) was applied to the wound sites once per day over five days postsurgery and then treated with QG at 2-day intervals until sacrifice. For the GPSMA group, each sample was applied on a wound site after the wound was created, and the nanofibers were exchanged at 2-day intervals until sacrifice. All of the wounds were covered with commercially available Tegaderm transparent film (3 M company, Maplewood, MN, USA) to keep the administered wound treatments in contact with the wound sites. The healing of each wound on the backs of the mice in all of the experimental groups was observed and photographed at days 0, 3, 8, 10, and 13. For each wound, the wound area was measured using ImageJ software, and the wound recovery was calculated. Wound recovery (%) = (A_0_ − A_i_)/A_0_ × 100, where A_0_ is the wound area at day 0, and A_i_ is the wound area at the predetermined time point.

### 4.7. Analysis of Development and Inflammation of Granulation Tissue during Wound Healing

An inflammation study was carried out on the blood plasma of the mice. Each blood sample was collected 1 day postsurgery and centrifuged for 20 min at 3000 rpm at 4 °C. The concentration of IL-6 in the blood plasma was determined by ELISA using an IL-6 ELISA kit (Biolegend, San Diego, CA, USA). On day 5 or day 8 after surgery, mice were sacrificed, and the wound tissue was excised carefully. The harvested wound tissues were bisected, with half subjected to a histological analysis and the other half frozen and stored at −80 °C for Western blot assay. The tissues were fixed with 4% formaldehyde, dehydrated in graded ethanol, embedded in paraffin, serially sectioned into 4-um-thick slides using a microtome, and stained with hematoxylin and eosin (H&E) or Masson’s trichrome. To detect the expression of CD31 (GeneTex, GTX130274), collagen I (GeneTex, GTX41286), collagen III (GeneTex, GTX111643), and alpha-smooth muscle actin (α-SMA, GeneTex, GTX100034) in the wound tissue, the samples were lysed with RIPA buffer with halt protease and a phosphatase inhibitor cocktail, shaken for 2 h at 4 °C, and centrifuged at 12,000 rpm for 20 min, after which the supernatants were collected. Samples containing equal amounts of protein were loaded onto commercial gradient SDS-PAGE gels. The other procedures were similar to those mentioned above. Beta-actin was used as a loading control, while the primary antibodies used contained CD31, collagen I, collagen III, and α-SMA.

### 4.8. Statistical Analyses

GraphPad software (v7.02, GraphPad Software, San Diego, CA, USA) was used for statistical analyses, and statistical comparisons were calculated using a one-way ANOVA test with Tukey’s test and an unpaired, two-tailed Student’s *t*-test. All data are presented as the mean ± standard deviation (SD), and *p*-values of less than 0.05 are considered statistically significant.

## 5. Conclusions

Wound care is a complicated process that blends art and science. As such, it takes considerable clinical experience to recognize and understand which treatment method is most appropriate for wounds with differing statuses. At present, it is difficult to find a wound dressing that can be applied throughout an entire wound healing process. This study, therefore, developed a new biomaterial that can be used in two forms depending on the particular goals of the wound healing treatment. QG can act as an endotoxin antagonist to reduce inflammation and prevent a chronic wound from occurring, after which the wound bed will transition from the inflammation phase to the proliferation phase. Meanwhile, GPSMA, another form of QG, is a wound dressing-like form that can remain for a long time at an injury area while providing oxygen by consuming hydrogen peroxide. The oxygen supply of a wound plays an essential role in the angiogenesis and collagen deposition required to reconstruct granulation tissue. To the best of our knowledge, therefore, QG is effectively a dual function wound dressing and has great potential for use as a positive advanced wound dressing in a wide range of applications, including the treatment of infected wounds as well as chronic nonhealing wounds.

## Figures and Tables

**Figure 1 ijms-21-01699-f001:**
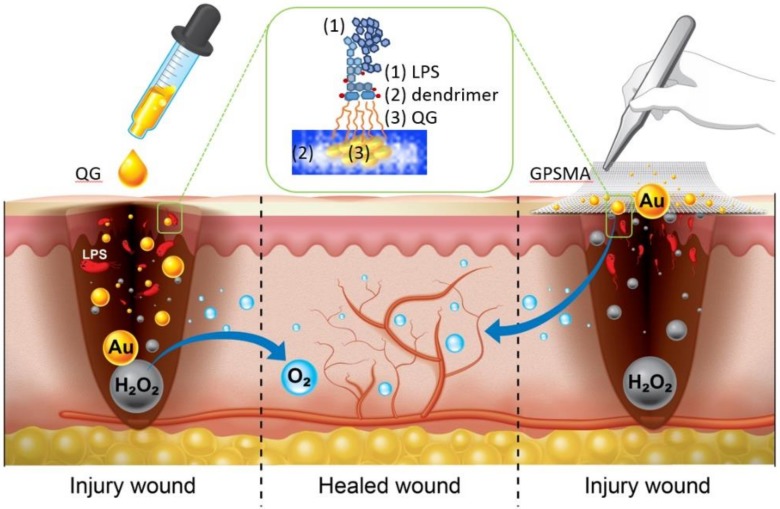
Schematic illustration of the impacts of a liquid form of quantized gold (QG) and a dressing form consisting of QG-grafted PSMA nanofibers (GPSMA) for the promotion of wound healing. The green rectangle shows the adhesive mechanism between QG and lipopolysaccharides (LPSs).

**Figure 2 ijms-21-01699-f002:**
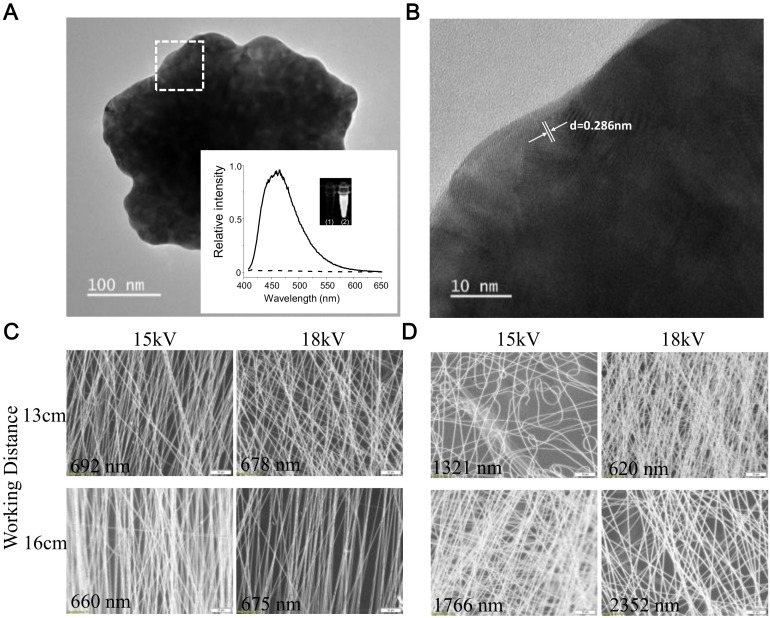
TEM and optical images of QG particles and nanofibers. (**A**) Low-magnification and (**B**) high-resolution TEM images (square region) showing a QG particle. The inset in (**A**) shows the fluorescence characterization of the QG particle. (**C**) 10 wt% and (**D**) 13 wt% poly(styrene-alt-maleic anhydride) nanofibers produced via the tuning of various electrospinning parameters. Scale bar: 50 m.

**Figure 3 ijms-21-01699-f003:**
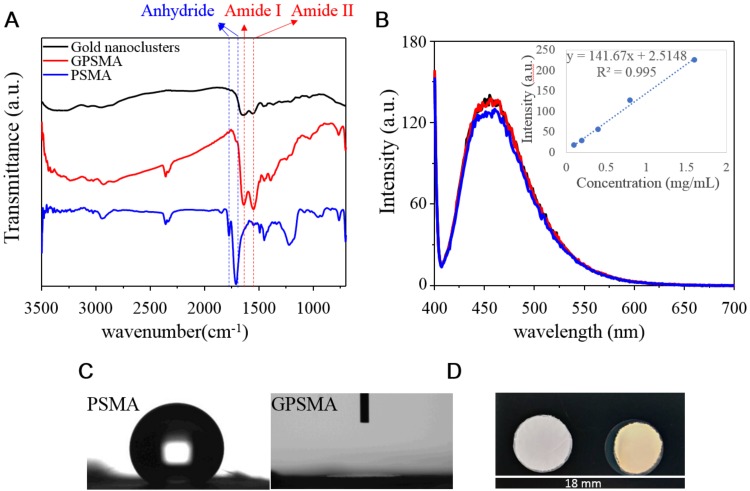
Assessments of the physicochemical properties of GPSMA. (**A**) FTIR spectra of QG, PSMA, and GPSMA; and (**B**) fluorescence spectra of free QG in aqueous solution after modifying the PSMA, with the measurements performed using an excitation wavelength at 390 nm. (**C**) Water contact angle measurements and (**D**) a macroscopic view showing the exterior changes of PSMA and GPSMA.

**Figure 4 ijms-21-01699-f004:**
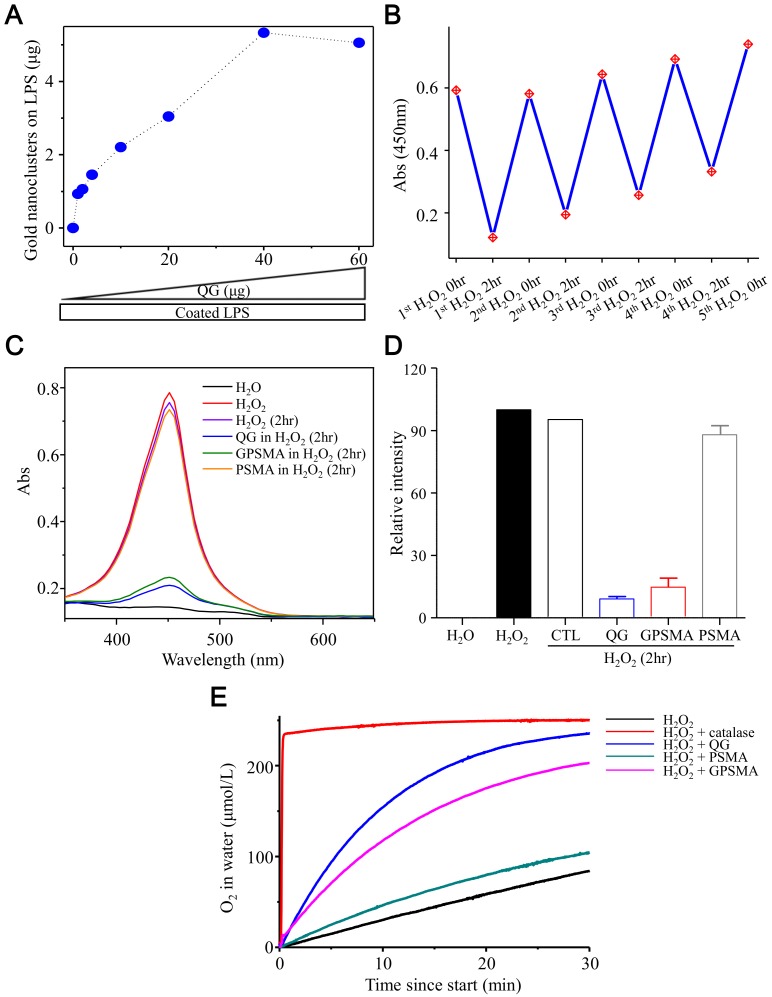
Catalytic assessment of QG and GPSMA. (**A**) Binding of varying amounts of QG to LPSs coated on well plates. (**B**) Potential recycling diagram of QG for catalyzing hydrogen peroxide. (**C**) Examination results of the catalase-like activity of QG, PSMA, and GPSMA using fluorescence spectroscopy. (**D**) The relative absorption intensities of catalase-like activity at 450 nm in each group are shown. H_2_O and H_2_O_2_ in their initial state were the negative control and positive control, respectively. (**E**) The measurement results of oxygen-generating kinetics in water after adding 0.1 mM H_2_O_2_ to each group.

**Figure 5 ijms-21-01699-f005:**
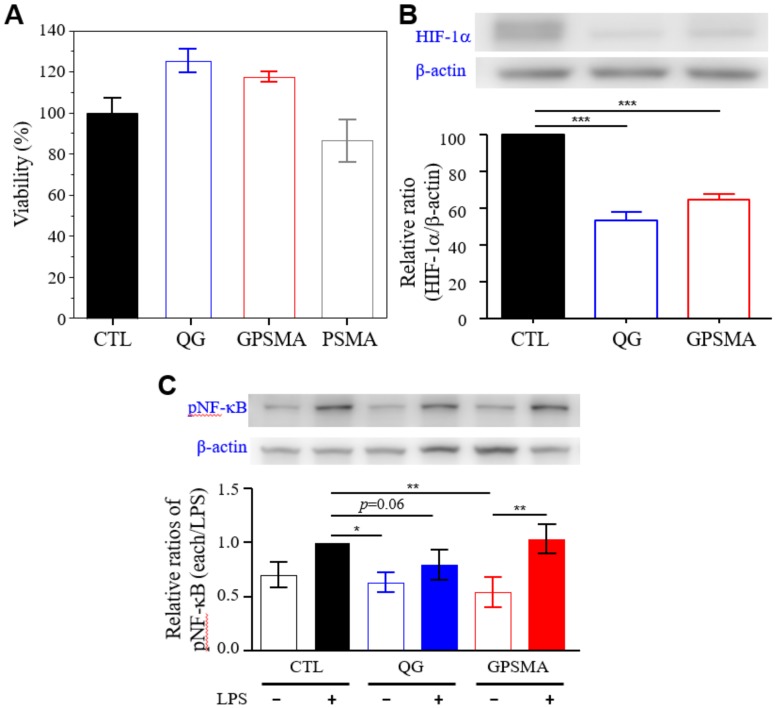
In vitro assessments of QG and GPSMA. (**A**) The viability of cells treated with QG, PSMA, and GPSMA in normoxic environments for 24 h. (**B**) Western blot analyses for the protein expression of HIF-1α in dermal fibroblasts under hypoxic conditions after being cocultured for 1 day (*n* = 3) and (**C**) the relative pNF-κB expression in RAW 264.7 cells in the absence or presence of LPSs. Both dermal fibroblasts and RAW 264.7 cells were treated with QG and GPSMA. Data are reported as the mean ± SD (*n* = 3); * indicates a significant difference between samples (* *p* < 0.05; ** *p* < 0.01; *** *p* < 0.001) using one-way ANOVA with Tukey’s multiple comparisons test. The *p*-value of QG + LPSs compared to LPSs was 0.0647, according to an unpaired test.

**Figure 6 ijms-21-01699-f006:**
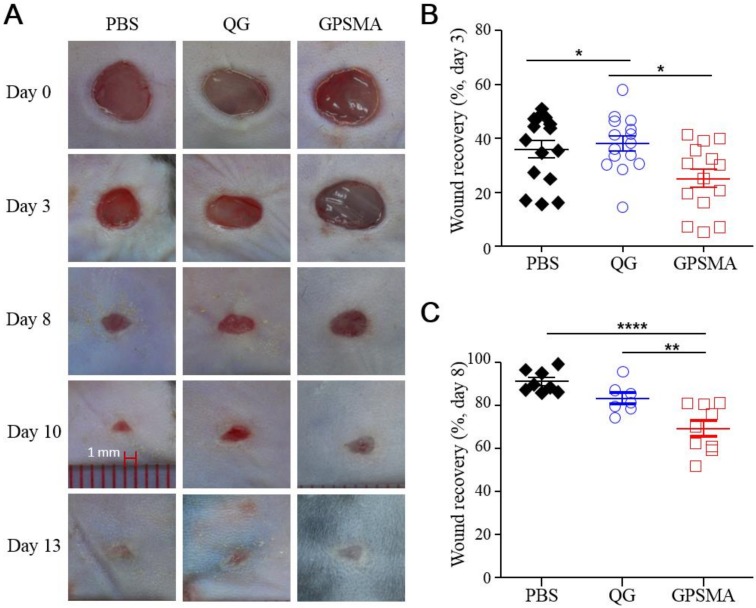
In vivo wound healing effects. (**A**) Representative digital images of wounds; (**B**) and (**C**) quantitative evaluation results of wound closure on the dorsal skin of mice treated with phosphate-buffered saline (PBS), QG, and GPSMA. Data are reported as the mean ± SD; * indicates a significant difference between samples (* *p* < 0.05; ** *p* < 0.01; **** *p* < 0.0001) using a one-way ANOVA with Tukey’s multiple comparisons test.

**Figure 7 ijms-21-01699-f007:**
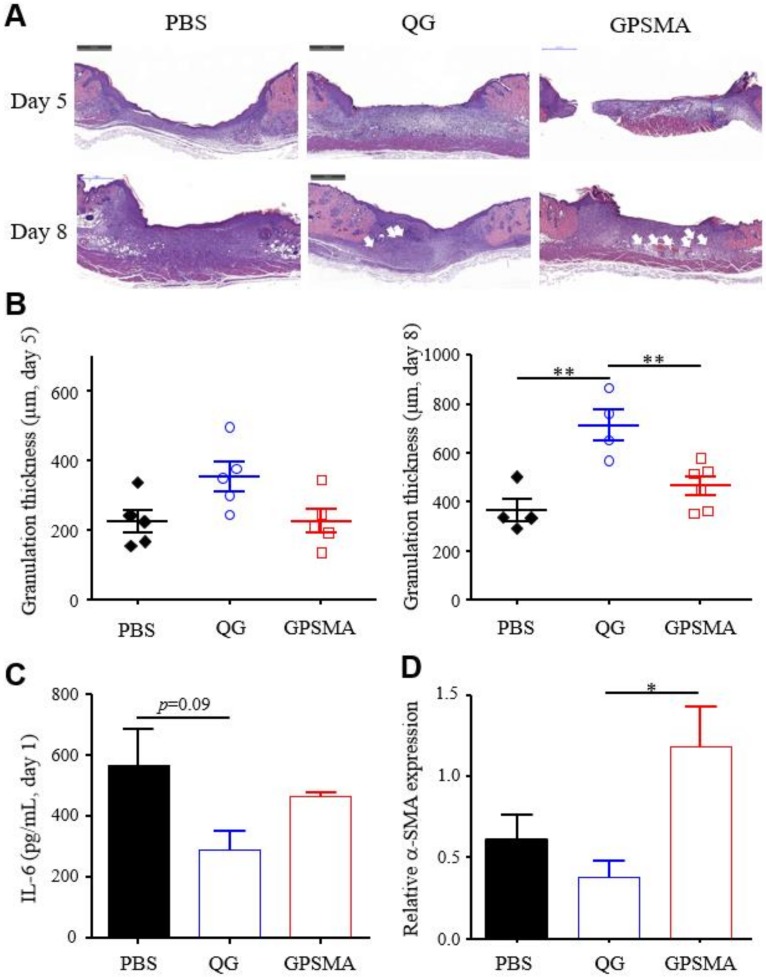
QG improved granulation growth and decreased the inflammation in wound beds. (**A**) A histological section stained with H&E (the scale bar is 500 μm) and (**B**) quantitative analysis results of granulation thickness. (**C**) The IL-6 levels as analyzed by ELISA and (**D**) the α-SMA levels as analyzed by Western blot analysis. The white arrows indicate blood vessels. Data are reported as the mean ± SD; * indicates a significant difference between samples (* *p* < 0.05; ** *p* < 0.01) using a one-way ANOVA with Tukey’s multiple comparisons test.

**Figure 8 ijms-21-01699-f008:**
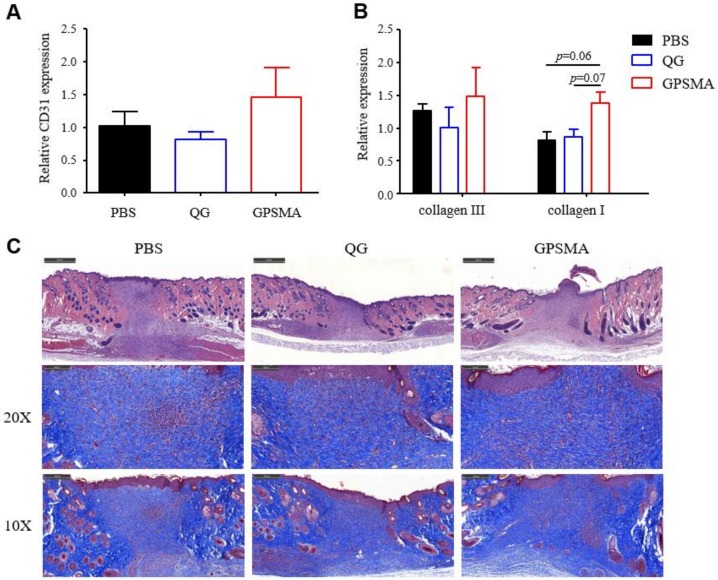
The effects of QG and GPSMA on granulation development. (**A**) The CD31, (**B**) collagen I, and collagen III levels were analyzed by Western blot analysis. Data are reported as the mean ± SD. In terms of the collagen I level, the *p*-values were 0.06 and 0.07 for the GPMSA group compared to the PBS group and the QG group using unpaired tests, respectively. (**C**) Histological section taken on day 13 stained with H&E and Masson’s trichrome. The collagen is stained blue, and the muscle tissue is stained red. Scale bar: 500 μm.

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
