# Peer review of "The Bioactive Core and Corona Synergism of Quantized Gold Enables Slowed Inflammation and Increased Tissue Regeneration in Wound Hypoxia"

_ijms, 2020, doi:10.3390/ijms21051699_

Round 1

Reviewer 1 Report

The manuscript entitled “The bioactive core and corona synergism of quantized gold enables slowed inflammation and increased tissue regeneration in wound hypoxia” explains a quantized gold (QG) and GPSMA systems to reduce the inflammation at the wound site and accelerate the wound healing by increasing local concentration of oxygen. Though the work looks interesting, there are several major concerns mentioned below to be addressed.

The authors should try to give a representative TEM picture of a group of nanoparticles to confirm their size range. Binding of QG to LPS should be characterized and explained. Moreover, it should also be checked whether binding of LPS to the QG surface has any effect on the formation of a stable biomolecular corona. In Fig. 5A, the presence of gold increased the cell viability or proliferation of primary dermal fibroblasts. The authors should explain the reason for this effect. HIF-1α protein bands in Fig. 5B are very faint and not clear. This experiment needs to be repeated for clear protein bands. 5C does not show any statistical difference in NF-kB expression between control, QG and GPSMA treatments. How can the authors claim that QG dramatically decreased the intracellular NF-kB levels? In Fig.6 QG and GPMSA groups low extent of wound closure even compared to the PBS group. This looks contradictory to their concept. How do the authors explain this? Moreover, in Fig. 7C there is no statistically significant difference shown in IL-6 expression of QG and GPSMA treatment compared to PBS. Also, it is surprising to see lower SMA expression in QG group compared to PBS, which should be explained. 8A and B also show no statistical significance, which might infer no difference in CD31 and collagen expression between the treatment groups. The authors should also study the release of nanoparticles from GPSMA. Were the concentrations of nanoparticles maintained same for QG and GPSMA groups wound healing experiments for comparison studies?

Author Response

Reviewer #1

Comments and Suggestions for Authors

  1. The manuscript entitled “The bioactive core and corona synergism of quantized gold enables slowed inflammation and increased tissue regeneration in wound hypoxia” explains a quantized gold (QG) and GPSMA systems to reduce the inflammation at the wound site and accelerate the wound healing by increasing local concentration of oxygen. Though the work looks interesting, there are several major concerns mentioned below to be addressed.

REPLY: Thank you for your comments and for giving us the opportunity to address your concerns on a point-by-point basis.

  1. The authors should try to give a representative TEM picture of a group of nanoparticles to confirm their size range. Binding of QG to LPS should be characterized and explained. Moreover, it should also be checked whether binding of LPS to the QG surface has any effect on the formation of a stable biomolecular corona.

REPLY: This is very import question. The structures of QGs and mechanism between QGs and LPS are studies and published in our previous work (Nano Lett. 2018, 18, 2864-2869.)

(1) The TEM images as well as showing in Figure 2A (current manuscript) that was taken by a TEM measurement at 200 kV was not clear enough to find a single QGs. Again, we also have known that QGs prefers to a layer-by-layer structure (Figure 2A) to form a film on TEM copper grid. In past, we have followed a work published by professor Dickson (J Am Chem Soc, 2003, 125, 7780) in order to use mass spectroscopy to answer the question. As expected, the mass spectra, as published in the supporting Figure S5 of our previous work (Chem Commun, 2010, 46, 2626-2628.), validated the presence of one dendrimer-encapsulated Au8-dominating nanoclusters (i.e., QGs comprised eight gold atoms within a single dendrimer). A detailed description of such composites has now been included in the main text (section 2.1) of this current work. Accordingly, we can confirm the formation of one dendrimer-encapsulated Au8-dominated nanoclusters as a main product of QGs. However, we cannot exclude the possibility of dendrimers having different QGs. Collectively, the peak of emission wavelength at 460 (Figure 2A inset) also confirms the Au8-dominated nanoclusters as a main product.

(2) With regard to the size estimation of the QGs, we tried to use a low-temperature STEM to count the number of gold atoms within the QGs. However, the organic layer (i.e., dendrimer) could effectively act as a shield that blocks such measurement. On the other hand, if we removed the organic layer, the removal process could destroy the stability of the QGs, resulting in a large-size nanoparticle formation. Again, the applicability of STEM for counting the number of gold atoms is limited to Au13 (J Am Chem Soc, 2007, 129, 12932), such that it could not be applied in the current study (i.e., in the case of Au8-dominated nanoclusters). Instead, we used a HRTEM with the highest electron power at 300 kV, which allowed clear TEM images to be taken (for example, Figure 2B in the current manuscript). However, the HRTEM can only capture QGs with layer-by-layer stacking in our case. Even though the real shapes of single QGs cannot be observed using currently available technology, the observation of thin-films also can explain that the geometry of QGs consists of a sheet-like structure that can allow layer-by-layer alignment. Until now, the atomic resolution of gold nanoclusters, especially those with sizes smaller than 1 nm, has posed a very challenging task (Science, 2014, 909, ref 18 of current manuscript). Here, we report QGs with layer-by-layer stacking with an atomic resolution, which is very important for the development of QGs. Given such QGs, we show that adhesive layers of dendrimers can cause the docking of lipid A of LPS (as showing in new Figure 1).

  1. In Fig. 5A, the presence of gold increased the cell viability or proliferation of primary dermal fibroblasts. The authors should explain the reason for this effect.

REPLY: We know that QGs can behave as enzyme mimics to show a catalase-like activity in our previous work (Small 2016, 23, 4127–4135; Small, 2017, 13, 1700278.). Currently, we cannot exclude QGs as scavengers or antioxidants which typically can eliminate reactive oxygen species (ROS) to increase the cell viability.

  1. HIF-1α protein bands in Fig. 5B are very faint and not clear. This experiment needs to be repeated for clear protein bands.

REPLY: Thank you for your comments. HIF-1α protein by western blotting have been repeated three times, please see the below three images (including 5B as the representative image). We also added n=3 in the figure legend.

  1. 5C does not show any statistical difference in NF-kB expression between control, QG and GPSMA treatments. How can the authors claim that QG dramatically decreased the intracellular NF-kB levels?

REPLY: We have modified our manuscript according to reviewer’s suggestion. We changed dramatically to slightly in the manuscript, and We also updated Fig. 5C for showing the p value.

  1. In Fig.6 QG and GPMSA groups low extent of wound closure even compared to the PBS group. This looks contradictory to their concept. How do the authors explain this?

REPLY: Thank you for your comments. At post-treatment day 3, QG group had faster wound closure than the PBS group (Fig. 6B), but faster closure of wound is not indicated well tissue regeneration in wound site. Thus, we further measured the amount of new granulation tissue and re-epithelialization formation by HE staining (Fig. 7A and 7B), angiogenesis by CD31 expression (Fig. 8A), and collagen deposition (Fig. 8B). Our results showed that QG and GPMSA groups had better tissue regeneration than the PBS group.

  1. Moreover, in Fig. 7C there is no statistically significant difference shown in IL-6 expression of QG and GPSMA treatment compared to PBS.

REPLY: Thank you for your comments. Wound healing involves a complicated pathophysiological process including inflammation, proliferation, and remodeling. And, effective control of inflammation in the wound bed is a crucial and necessary aspect of effective wound healing treatments. We rechecked the statistics analysis of Fig. 7C. Analytic Results showed that here were slightly lower blood IL-6 levels between the QG group and the PBS group (p=0.09). We also update Fig.7C and showed the p value in the manuscript. Thus, providing an appropriate supply of QG is the best method for shortening the inflammation phase of tissue regeneration, which in turn accelerates the next phase of wound healing process. These IL-6 levels of the QG group (on day1) were consistent with faster wound recovery of the QG group at post-treatment day 3 (Fig. 6B). By contrast, although the anti-inflammatory effects of the GPSMA were a little less strong than those of the QG, the GPSMA could maintain a little inflammation by catch the amount of LPS within the wound bed and the GPSMA provided oxygen generation within the wound bed to help well tissue regeneration.

  1. Also, it is surprising to see lower SMA expression in QG group compared to PBS, which should be explained.

REPLY: Wound recovery of the QG group at post-treatment day 8 was comparable to that of the PBS group, indicating that the phase of wound healing of both groups were coming to the end of wound healing. α-SMA, a key marker of myofibroblasts involved in wound contraction and scar formation (ref33 Chitturi et al., 2015). And the apoptosis of myofibroblasts initiates at the end of the wound healing and re-epithelialization( (1)Am J Pathol. 1995 Jan; 146(1):56-66. Apoptosis mediates the decrease in cellularity during the transition between granulation tissue and scar.   (2)Journal of Investigative Dermatology, Volume 127, Issue 3, March 2007, Pages 526-537, Formation and Function of the Myofibroblast during Tissue Repair). Thus, the appearance of lower α-SMA expression between QG and PBS groups was reasonable which was consistent with the results of wound closure between QG and PBS group.

  1. 8A and B also show no statistical significance, which might infer no difference in CD31 and collagen expression between the treatment groups.

REPLY: We have updated Fig. 8B for showing the p value. The difference between the QG and the GPSMA is that the wound dressing form of the GPSMA can be retained long-term at a wound site. GPSMA slightly makes it possible QG on the wound dressing to produce oxygen generation in the local injury area, in turn to help angiogenesis not by HIF-1a-independent signal. This research focused on the effects on wound healing process by liquid form or wound dressing form of QG, which has catalase-like activity. The data revealed that both of QG forms positively affected on local wound site. Therefore, we believed that QG could be used on wound by selecting the QG type depending on the purpose of wound healing (shoring the inflammation or assisting tissue regeneration).

  1. The authors should also study the release of nanoparticles from GPSMA. Were the concentrations of nanoparticles maintained same for QG and GPSMA groups wound healing experiments for comparison studies?

REPLY: Yes, we have checked that the catalase-like activity can still perform from the used GPSMA, which indicated that the QGs still attached in the nanofiber surface. Again, the concentration of QGs in nanofiber was about 1.2mg/patch (please see the calibration curve in Figure 3B), which is comparable to liquid QGs

Reviewer 2 Report

The authors developed a nanozyme that gold atoms entrapped within a dendrimer, which can eliminate endotoxin to shorten inflammation stage. Moreover, the nano-gold can decompose the endogenous H2O2 into O2, so the treatment can induce wound regeneration. In addition, the two form of efficacy, a liquid form and a dressing form, provide more applications on wound healing. However, a few of statements were unclear and more discussion was necessary to support their claims. I recommend minor revision as noted below.

  • First, the PAMAM on the surface of nano-gold binds with lipopolysaccharide (LPS) to eliminate endotoxin. However, Figure 1 only shows the schematic illustration of the catalyzing ability of quantized gold (QG). The authors should modify the scheme to describe the structure of QG and mention the mechanism of LPS elimination by the PAMAM.

  • In Figure 3C & D, the authors displayed the morphology of the nanofibers fabricated with various electrospinning parameters, including working distance, voltage and concentration of PSMA solution. They also mentioned “ PSMA nanofibers with diameters of 660 nm could be produced with a 10 wt% polymeric solution, 15 kV voltage, 1 mL/h feeding rate, and 16 cm working distance.” in the main text. The case seems the optimal parameters, but no reason and discussion were provided. The authors should explain the benefits of that parameter for further experiments and applications.

  • The FTIR spectra of QG, PSMA, and GPSMA in Figure 3A. The statement of the peaks of amide Ⅰ and Ⅱ as well as anhydride group apparently disagreed with those found from the FTIR spectra in Figure 3A. Please correct the FTIR peaks.

  • In Figure 4, the authors evaluated the catalytic ability of QG. Interestingly, the catalytic ability of artificial enzymes, especially for noble metal, strongly differs with the pH value of the environment. In this study, the authors only assessed their catalyst in water without any pH control. I recommend the authors should conduct more experiments at various pH values, especially considering the wound-like microenvironment.

  • The scale bars in the figures of 2C, 2D, 3D, 6A, 7A, and 8C are missing or unclear. Please modify the figures and corresponding captions.

Author Response

Reviewer #2

Comments and Suggestions for Authors

The authors developed a nanozyme that gold atoms entrapped within a dendrimer, which can eliminate endotoxin to shorten inflammation stage. Moreover, the nano-gold can decompose the endogenous H2O2 into O2, so the treatment can induce wound regeneration. In addition, the two form of efficacy, a liquid form and a dressing form, provide more applications on wound healing. However, a few of statements were unclear and more discussion was necessary to support their claims. I recommend minor revision as noted below.

  1. First, the PAMAM on the surface of nano-gold binds with lipopolysaccharide (LPS) to eliminate endotoxin. However, Figure 1 only shows the schematic illustration of the catalyzing ability of quantized gold (QG). The authors should modify the scheme to describe the structure of QG and mention the mechanism of LPS elimination by the PAMAM.

REPLY: We provided the Figure 1 with the mechanism of LPS elimination by the PAMAM. The structures of QGs and mechanism between QGs and LPS are studies and published in our previous work (Nano Lett. 2018, 18, 2864-2869.). We also report QGs with layer-by-layer stacking with an atomic resolution, which is very important for the development of QGs. Given such QGs, we show that adhesive layers of dendrimers can cause the docking of lipid A of LPS (as showing in new Figure 1).

  1. In Figure 2C & D, the authors displayed the morphology of the nanofibers fabricated with various electrospinning parameters, including working distance, voltage and concentration of PSMA solution. They also mentioned “ PSMA nanofibers with diameters of 660 nm could be produced with a 10 wt% polymeric solution, 15 kV voltage, 1 mL/h feeding rate, and 16 cm working distance.” in the main text. The case seems the optimal parameters, but no reason and discussion were provided. The authors should explain the benefits of that parameter for further experiments and applications.

REPLY: Thanks for reviewer’s suggestion, the reason was provided in the main text.

……………………. The process used to from the PSMA nanofibers has the advantage that it is easy to generate a higher surface area by tuning the electrospinning parameters in order to obtain high grafting amounts of QG in the subsequent steps. In general, the smaller diameter of nanofibers exhibits the higher surface area. Comparing the Figure 2(c) and (d), PSMA nanofibers shows more stable structure in Figure 2(c). Because it can be stably obtained PSMA nanofibers in 10wt% of PSMA solution, and the higher concentration of PSMA solution (13 wt%) cannot be maintained and control during the electrospinning process. As shown in Figure 2(c) and (d), PSMA nanofibers with diameters of 660 nm could be produced with a 10 wt% polymeric solution, 15kV voltage, 1mL/h feeding rate, and 16 cm working distance which has the smallest diameter……………………………………….

  1. The FTIR spectra of QG, PSMA, and GPSMA in Figure 3A. The statement of the peaks of amide Ⅰ and Ⅱ as well as anhydride group apparently disagreed with those found from the FTIR spectra in Figure 3A. Please correct the FTIR peaks.

REPLY: We have correct the marked lines of FTIR spectra in Figure 3A.

  1. In Figure 4, the authors evaluated the catalytic ability of QG. Interestingly, the catalytic ability of artificial enzymes, especially for noble metal, strongly differs with the pH value of the environment. In this study, the authors only assessed their catalyst in water without any pH control. I recommend the authors should conduct more experiments at various pH values, especially considering the wound-like microenvironment.

REPLY: We have known that QGs can behave as enzyme mimics in the different pH values to show a catalase-like activity in our previous work (Small 2016, 23, 4127–4135; Small, 2017, 13, 1700278.). Our measurements specially show the catalase-like activity of QGs, which can caver the microenvironments of wound sites.

  1. The scale bars in the figures of 2C, 2D, 3D, 6A, 7A, and 8C are missing or unclear. Please modify the figures and corresponding captions.

REPLY: All scale bars have provided in Figure captions (2C, 2D, 7A and 8C) or Figures (3D, 6A).

Round 2

Reviewer 1 Report

I deeply appreciate the effort put by the authors in addressing the comments/concerns raised. However, the below comment needs to be addressed after which I recommend the manuscript for publication.

Comment:

  1. In response to the comment regarding the release of nanoparticles from GPSMA, it was mentioned that catalase-like activity can still perform from the used GPSMA, which they believe as a proof that the QGs are still attached in the nanofiber surface. The authors should clarify how the QGs are exposed to the wound surface when they are still attached and embedded in pores of PSMA nanofibers. It is also suggested that the authors should briefly include this hypothesis in the revised manuscript.

Author Response

2nd-round revision

Reviewer 1

Q: In response to the comment regarding the release of nanoparticles from GPSMA, it was mentioned that catalase-like activity can still perform from the used GPSMA, which they believe as a proof that the QGs are still attached in the nanofiber surface. The authors should clarify how the QGs are exposed to the wound surface when they are still attached and embedded in pores of PSMA nanofibers. It is also suggested that the authors should briefly include this hypothesis in the revised manuscript.

REPLY: In the section of material and method, we have actually described the detailed the conjugation method of QGs and PSMA. The post-modification method can guarantee QGs to exert the catalase-like activity. As your concern, we have validated the function of each batch. If used only blend the QGs and PSMA, the catalase-like activity of QGs cannot be performed due to QGs can be embedded within PSMA. In our finding, thus, the post-modification of QGs is key to keep the surface accessibility of QGs for catalyze the decomposition of hydrogen peroxide.